# Accuracy and Sheet Thinning Improvement of Deep Titanium Alloy Part with Warm Incremental Sheet-Forming Process

**Badreddine Saidi** [1,2] **, Laurence Giraud Moreau** [3,*] **, Abel Cherouat** [4] **and Rachid Nasri** [1]

1 Applied Mechanics and Engineering Laboratory (LR-11-ES19), University of Tunis El Manar, ENIT, Le Belvédère, Tunis BP 37-1002, Tunisia; badreddine.saidi@enit.utm.tn (B.S.); rachidnasri2003@yahoo.fr (R.N.)
2 Higher Institute of Technological Studies of Rades, Rades BP 172-2098, Tunisia
3 UR-LASMIS, University of Technology of Troyes, 12 rue Marie Curie, 10004 Troyes, France
4 UR-GAMMA3, University of Technology of Troyes, 12 rue Marie Curie, 10004 Troyes, France; abel.cherouat@utt.fr
* Correspondence: laurence.moreau@utt.fr; Tel.: +33(0)-3-25-71-80-42

**Abstract:** Incremental forming is a recent forming process that allows a sheet to be locally deformed with a hemispherical tool in order to gradually shape it. Despite good lubrication between the sheet and the tip of the smooth hemisphere tool, ductility often occurs, limiting the formability of titanium alloys due to the geometrical inaccuracy of the parts and the inability to form parts with a large depth and wall angle. Several technical solutions are proposed in the literature to increase the working temperature, allowing improvement in the titanium alloys' formability and reducing the sheet thinning, plastic instability, and failure localization. An experimental procedure and numerical simulation were performed in this study to improve the warm single-point incremental sheet forming of a deep truncated cone in Ti-6Al-4V titanium alloy based on the use of heating cartridges. The effect of the depth part (two experiments with a truncated cone having a depth of 40 and 60 mm) at hot temperature (440 °C) on the thickness distribution and sheet shape accuracy are performed. Results show that the formability is significantly improved with the heating to produce a deep part. Small errors are observed between experimental and theoretical profiles. Moreover, errors between experimental and numerical displacements are less than 6%, which shows that the Finite Element (FE) model gives accurate predictions for titanium alloy deep truncated cones.

**Keywords:** warm single-point incremental forming (SPIF); truncated cone; temperature effect; FE analysis

## 1. Introduction

Many small parts with complex shapes must be produced in small series, thus making the cost of the equipment proportionately very large in relation to the overall cost of the part. Thus, the development of processes with low production costs seems interesting for the production of small series parts or the manufacturing of prototypes. The incremental sheet-forming process is an emerging process, which allows producing asymmetrical complex parts for various application in transportation and biomedical. The incremental forming process presents an alternative to the conventional sheet metal-forming processes such as stamping and drawing [1]. It is based on the use of a spherical punch, which moves along Computer Numerical Control (CNC) or robots-controlled trajectories similar to those of machining [2] and useful to manufacture customized products [3]. The use of robot arms can significantly reduce the manufacturing time and it is possible to deform, bend, flange, and load/unload the part with the same fixture [4].

Several techniques have been developed to decrease the shape errors in the conventional incremental forming process: warm incremental forming technology, optimization of the process parameters, tool path optimization, use of circumferential hammering tool,

etc. Depending on the number of contact points between the sheet and punch used, incremental sheet forming may be divided into two main categories processes to produce mass customized, double-curved, three-dimensional forms from sheet metal:

- Single-Point Incremental Forming (SPIF) is done using a single tool and no need for tailored tools and dies [4–6]. The major parameters in SPIF have been identified, and process capabilities are being expanded [7,8]. An incremental sheet-forming process has been successfully employed at room temperature for many sheet metals especially those with good formability such as aluminum alloy [9], stainless steel [10] and magnesium sheets [11].
- Two-Point Incremental Forming (TPIF) is similar to SPIF, but on the other side of the sheet is a local supported partially by a die to ensure a better and more precise shape of the final part [3].

The Ti6Al4V titanium alloy is one of the most important engineering alloys, combining attractive properties with inherent workability, excellent combination of high specific strength (strength/weight ratio), fracture-resistant characteristics, and exceptional resistance to corrosion for the latter, allowing it to be fabricated into complex shapes and medicals parts [12]. In case of biomedical work pieces characterized with a complex shape and high depth, single-point incremental forming (SPIF) is usually carried out, but the Ti6Al4V titanium alloy limited formability at room temperature forces their conduction at an elevated temperature, which allows also reducing spring-back and, therefore, increasing the part geometrical accuracy [13]. The specific use of temperature as a process parameter opens the possibility of improving the formability of Ti6Al4V titanium alloy and of achieving a significant reduction in the forces or pressures required for forming. More recently, the selective laser sintering (SLS) additive manufacturing technologies is widely employed to produce complex parts made of Ti6Al4V powder [14,15].

In recent years, different solutions were proposed in order to increase the formability of the Ti6Al4V alloy [16]. Several results have shown that increasing the working temperature has many benefits in improving the formability and reducing the forming force [3]:

1.  Göttmann et al. [17] proposed a new machine setup that projects an elliptical laser onto Ti-A16V4 sheets in order to attempt a heating of the forming zone at 400 °C. Other solutions were proposed to improve the titanium formability titanium alloy.
2.  Xu et al. [18] combined high-speed rotation of the punch tool with an electric static heating to reach a high-level temperature (400 °C) in SPIF that is enough to affect the material behavior.
3.  Honarpisheh et al. [19] investigate experimentally and numerically electric hot incremental forming and study the effect of process parameters (wall angle, step size, and tool diameter) on the forming force and thickness distribution of the final part.
4.  Vahdani et al. investigated using resistance as an electric current to generate a heat SPIF setup for the contact zone between the forming tool and the sheet [20]. The obtained results show that the formability of the Ti-6Al-4V alloy sheet strongly depends on the lubrication condition.
5.  Liu [21] presented a state-of-the-art review of heat-assisted incremental sheet forming. The author groups together the works carried out, in particular those used for heating titanium alloys, laser heat [17], friction heat [22], electric heat [20], induction heat [23], and combined heat friction [24].
6.  Jin et al. present several warm SPIF (WSPIF) methods to improve the formability and overcome the low geometrical accuracy [25].

Cartridge heaters are an excellent choice to use as a conductive source for heating the forming surface to a forming temperature or range of forming temperatures [10]. A cartridge heater is tube-shaped and can be inserted into drilled holes. The located cartridges are connectable to a source of electrical power such that when each heating cartridge is powered on for an identical period of heating time, a temperature distribution is produced

within the metal body portion that maintains the sheet material forming temperature at the forming surface.

In addition, the SPIF forming limit related to wall thinning is represented by a maximum wall angle. Increasing the maximum wall angle requires improvements to the SPIF process. The process can be improved by adding local heating, but other factors must also be taken into account.

In the present work, experimental and numerical analysis of a truncated cone deep warm WSPIF setup based on the use of cartridge heaters is investigated. Experimental tests are performed in order to optimize the placement of the cartridge heaters with an acceptable temperature distribution. The cartridge heaters are included in a lower blank holder in order to provide a uniform distribution of heat in the sheet [13]. The proposed cartridge heaters technique is adapted to a machining-controlled machine and monitoring system in order to measure the forming force and temperature. Using optimized process parameters such as the step size, tool diameter, and tool path [26], we investigate in this study the effect of forming the step depth of Ti-6Al-4V alloy at warm temperature between 300 and 700 °C on the thickness reduction distribution and truncated cone profile. Numerical simulation is proposed to investigate the thermo-mechanical behavior depth of Ti-6Al-4V alloy during the deep WSPIF forming process in order to predict the thickness distribution and the final profile shape [27].

## 2. Materials and Methods

### 2.1. Experimental Warm SPIF Setup

The sheet metal forming of Ti-6Al-4V alloy can be done at room temperature by local drawing operation, at very high temperature ($\simeq$900 °C) by superplastic deformation, and at intermediate temperature ($\geq$700 °C) by hot forming. In order to reduce the production costs, spring back, the machine working forces, and the cycle times reduction associated with a decrease in temperature levels are relevant.

The principle of warm incremental sheet-forming setup is the heating and maintaining of the sheet metal at working temperature. A hot-forming tool is heated with multiple electrical resistance cartridge heaters (see Figure 1). The heaters are located in the body of the tool to maintain the sheet within a predetermined temperature range. The sheet to be deformed rests on the die to have a contact surface, allowing it to be heated by conduction throughout the forming operation.

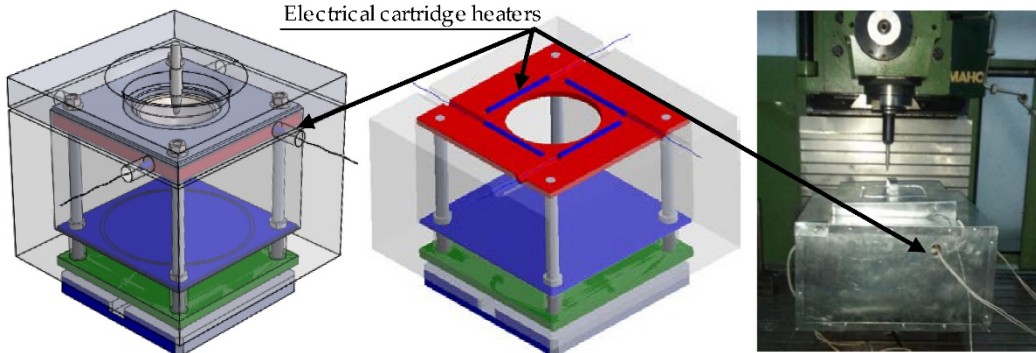

**Figure 1.** Warm incremental sheet-forming setup with cartridge heaters.

Experimental thermal and optimization analyses direct the placement of the heaters inside the lower sheet holder so that when each heating zone is powered, an acceptable temperature distribution will be produced within the holder operating temperature. The warm experimental setup of a single point incremental sheet forming presented in Figure 2 includes the following:

(a)　A multiaxial load cell force sensor FN7325-M6 provides monitoring and measurement of the forces. This sensor measures forces up to 5000 N and moments up to 200 Nm

on the X and Y axes, and along the Z axis, forces up to 250 kN and moments up to 7000 Nm. The working temperature range is between −20 and 80 °C, and for the reliability of the measurements, the sensor is calibrated.

(b) The incremental forming process is carried out with a 3-axis CNC vertical milling machine MAHO.

(c) The Ti-6Al-4V alloy sheet size (300 × 300 mm) with initial thickness $t_i$ = 0.5 mm and the effective working area was (70 × 70 mm) is formed with spherical punch controlled by computer machine. The diameter of the punch of dp = 5 mm is made of X160CrMoV12 steel, which has undergone a heat treatment (55HRC Hardness). The punch speed S = 50 rpm and Feed rate f = 600 mm/min are chosen in order to reduce the effect on the heat generation due to the friction with the sheet (see Figure 3). The incremental forming of Ti-6Al-4V alloy, the mechanical characteristics of which are given in Table 1, is carried out for a temperature in the range of 400 to 600 °C and a step down Δz of 0.5 mm.

(d) The real-time temperature monitoring is provided by an IRISYS 4000 type infrared camera.

(e) The reverse engineering approach for the surface reconstruction of CAD models starting from 3D mesh data is performed to analyze the formability of a deformed truncated cone with a wall angle α of 50°, using three-dimensional Coordinate-Measuring Machine.

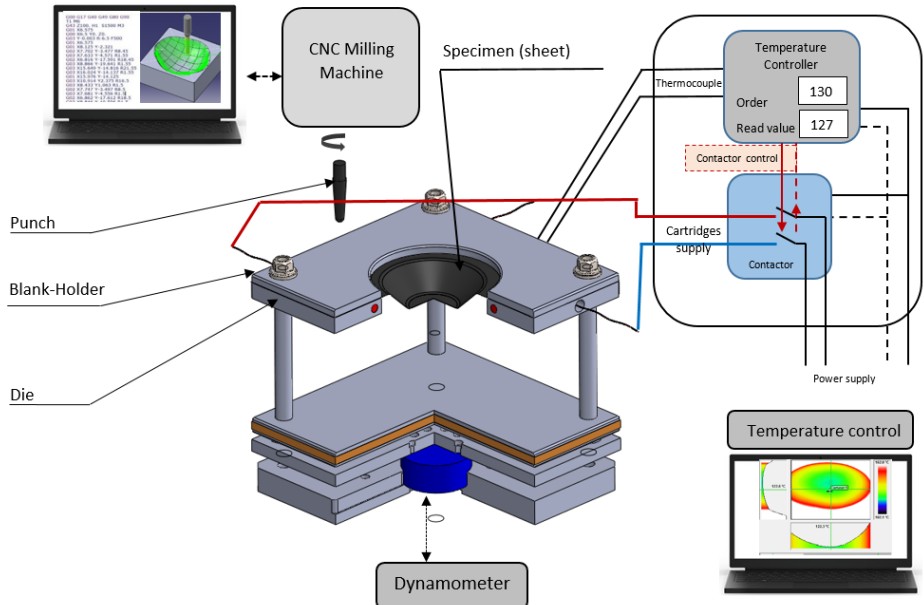

**Figure 2.** Warm incremental sheet-forming setup (WSPIF).

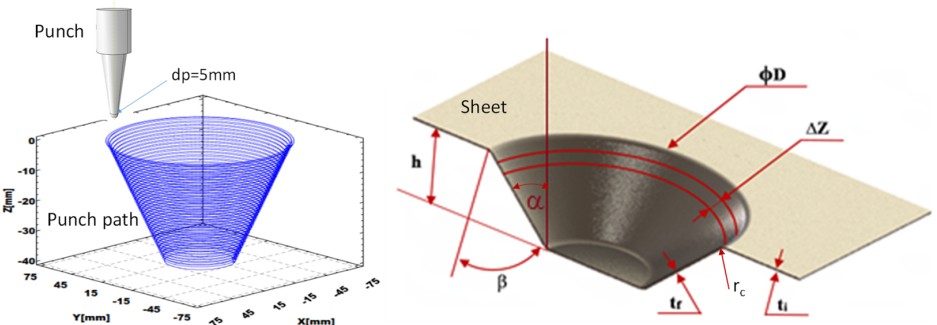

**Figure 3.** The shape CAD of the truncated cone used to create a punch toolpath.

**Table 1.** The incremental sheet-forming conditions.

| No. | $t_i$ [mm] | $r_c$ [mm] | $\alpha$ [°] | dp [mm] | $\Delta Z$ [mm] | ØD [mm] | S [rpm] | f [mm/min] | T [°C] | h [mm] |
|---|---|---|---|---|---|---|---|---|---|---|
| 1 | 0.5 | 5 | 50 | 5 | 0.5 | 130 | 50 | 600 | 450 | 40 |
| 2 | 0.5 | 5 | 50 | 5 | 0.5 | 130 | 50 | 600 | 450 | 60 |

### 2.2. Experimental Test Conditions

To choose the range of the different test parameters, mainly forming temperature and tool speeds, a corrective measure on the forming conditions is considered [28]. For a low temperature (300 °C), the sheet is not too much affected by the heating, and it does not move too far from the cold-forming conditions. We prove also that for a heating exceeding 700 °C, the blank sheet becomes very soft, and the tool tears the sheet instead of deforming it. Figure 4 presents the final truncated cone at different forming temperatures. A good compromise formability heating is provided for a temperature in the range of 400 and 600 °C. For the used titanium alloys sheet Ti-6Al-4V, the WSPIF tests for truncated cones with two punch depths (h = 40 mm and h = 60 mm) were carried out for the temperature of T = 450 °C with an uncertainty of 50 °C [29,30].

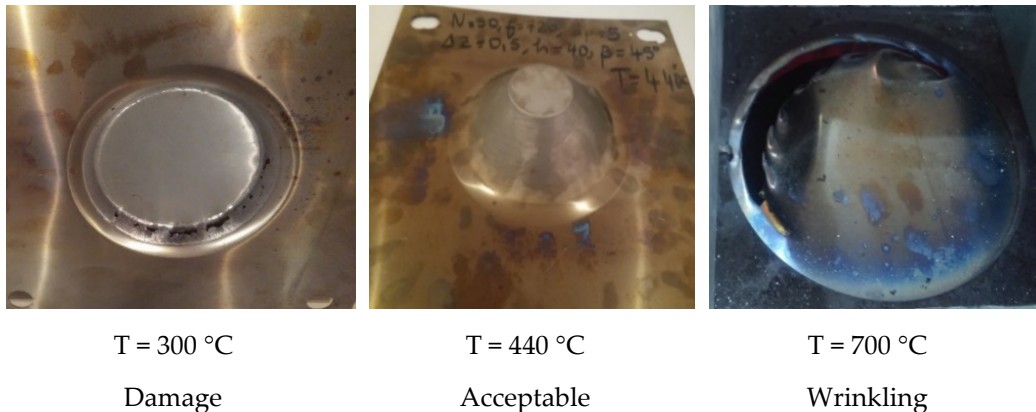

| T = 300 °C | T = 440 °C | T = 700 °C |
|---|---|---|
| Damage | Acceptable | Wrinkling |

**Figure 4.** Truncated cones with a wall angle of 45° formed at different temperature.

The optimized parameters presented in the Table 1 are chosen by varying only the depth of the truncated cone to determine the limit of formability through the thickness distribution and the profile error. The exploitation of the scan results is carried out via the development of a script to determine the thickness distribution and final profile shape of the truncated cone according to cutting plans. Figure 5 presents the Coordinate-Measuring Machine (Figure 5a) used to measure the profile (Figure 5b). FE analysis of the incremental forming was executed through the ABAQUS code under loads similar to those obtained from the experimental study.

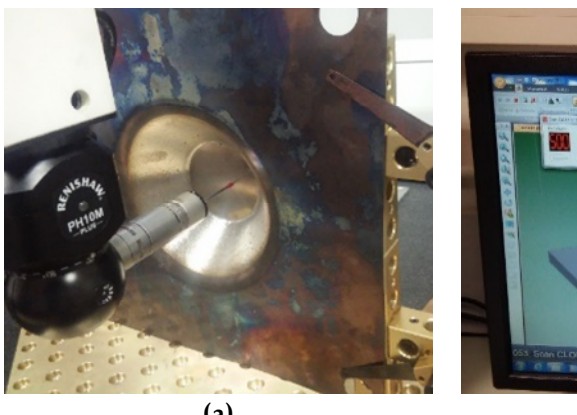 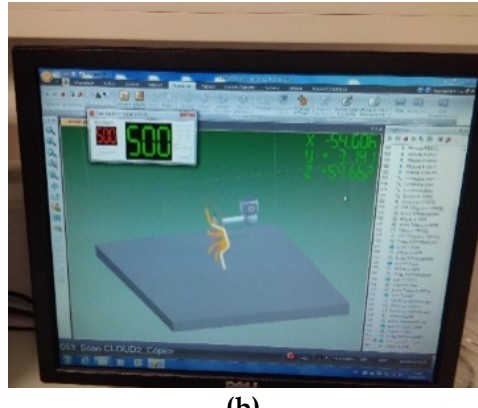

(a)                                                                                                                  (b)

**Figure 5.** Scan truncated cone (**a**) with three-dimensional measuring machine (MMT) (**b**).

*2.3. Finite Elements Modeling of Warm SPIF*

Numerical simulation of the warm incremental forming process was performed with ABAQUS FE software. The dynamic displacement temperature explicit solver was used to simulate the finite thermo-elasto-viscoplastic deformation with contact friction of thin titanium alloys sheet. Linear four-node thermally coupled stress/displacement shell elements (S4RT) with reduced integration and a large-strain formulation are used.

In order to reduce the CPU computational calculation, the mesh of the sheet was divided into three zone partitions (see Figure 6):

- A coarse mesh (480 finite elements with a size of 10 mm × 10 mm) in the large clamping zone 1.
- A circular fine mesh (1600 finite elements) in a useful zone 2 of 60 mm diameter (3 mm × 3 mm).
- Fine mesh (1200 finite elements) in the fine tip zone 3 (0.5 mm × 0.5 mm).

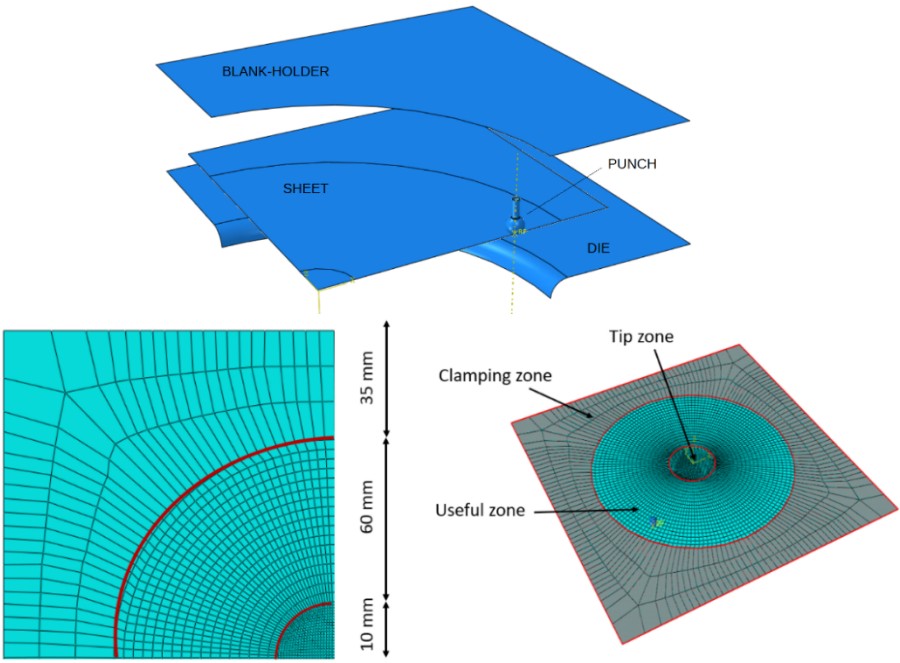

**Figure 6.** The 3D CAD model, FE meshing, and partition of initial blank.

The 3D rigid tool of diameter dp = 5 mm was modeled by an analytical rigid surface. During the experimental tests, the tool path had been created by CAD/CAM software. The tool path was the same one used during the experiments with warm SPIF. The interaction

between the blank and the spherical punch was defined by an isotropic Coulomb's friction model ($\mu = 0{,}1$) [27,29].

The thermo-viscoplastic material of the Ti-6Al-4V titanium alloy was defined by using the Johnson–Cook behavior model given the equivalent stress versus strain as [30]:

$$\sigma_y = (A + B.\varepsilon^n).\left[1 + C.ln\left(\frac{\dot{\varepsilon}}{\dot{\varepsilon_0}}\right)\right].\left[1 - \left(\frac{T - T_r}{T_m - T_r}\right)\right] \tag{1}$$

where ($\sigma_y$) is the yield stress, ($\varepsilon$) is the equivalent plastic strain, ($\dot{\varepsilon}$) is the equivalent plastic strain rate, ($\dot{\varepsilon_0}$) is the reference equivalent plastic strain rate, ($T$) is the work temperature, ($T_m$) is the melting temperature, and ($T_r$) is the room temperature.

The Johnson–Cook model was chosen because of its ability to take into account the temperature in the material modeling during the numerical simulation.

Johnson–Cook's thermos-viscoplastic parameters of the Ti-6Al-4V titanium alloy are given in Tables 2–4 (see [30–32]).

**Table 2.** Mechanical properties of the titanium alloys Ti-6Al-4V [30,31].

| T [°C] | Young's Modulus E [MPa] | Poisson's Ratio $\nu$ | Density $\rho$ [kg/m$^3$] |
|---|---|---|---|
| 21.11 | 117,210 | 0.31 | 4430 |
| 204.44 | 106,870 | 0.31 | 4430 |
| 426.67 | 95,150 | 0.31 | 4430 |
| 648.89 | 82,720 | 0.31 | 4430 |

**Table 3.** Thermal properties of the titanium alloys Ti-6Al-4V [30,31].

| Temperature [°C] | Heat Conductivity $\alpha$ [W/m/°C] | Expansion $\lambda$ [µm/m/°C] | Specific Heat Cp [J/kg/°C] |
|---|---|---|---|
| 17.78 | 6.92 | 1.13 | 387.56 |
| 93.34 | 7.44 | 1.13 | 406.93 |
| 204.44 | 8.65 | 1.13 | 426.31 |
| 426.67 | 11.94 | 1.13 | 474.76 |
| 537.78 | 13.67 | 1.13 | 517.39 |
| 958.22 | 18 | 1.13 | 697.61 |

**Table 4.** Jonson–Cook's parameters for the titanium alloys Ti-6Al-4V [32].

| A [MPa] | B [MPa] | C | m | n | $\dot{\varepsilon_0}[s^{-1}]$ | $T_m[°C]$ | $T_r[°C]$ |
|---|---|---|---|---|---|---|---|
| 928 | 1062 | 0.0167 | 0.75 | 0.62 | $10^{-3}$ | 1663 | 25 |

## 3. Results and Discussions

The simulation of the warm incremental forming process was performed with two values of truncated cone depth: the same values as those studied experimentally (h = 40 mm and h = 60 mm). In previous work [27], the results of the tests carried out with the WSPIF setup on a titanium alloy Ti-6Al-4V sheet at T = 450 °C show that it is possible to produce smooth seines truncated cones without fracture appearance for a wall angle not exceeding ($\alpha = 56°$) and depth (h = 40 mm).

To check the formability of titanium alloys Ti-6Al-4V in WSPIF, the maximum depth is reached during the forming of a deep truncated cone (Figure 3). We aim to conduct a

comparison between the experimental and numerical results on the thickness distribution and the final profile shape with deep depth.

In order to see the behavior of the Ti-6Al-4V titanium alloy at WSPIF and compare deep truncated cones via a forming program for deep truncated cones ($\alpha$ = 50°) and different depth parts (h = 40 and 60 mm), we used a punch diameter (dp = 5 mm) with an axial increment ($\Delta$z = 0.5 mm). This truncated cone work is experimentally performed and numerically simulated, as shown in Figure 7.

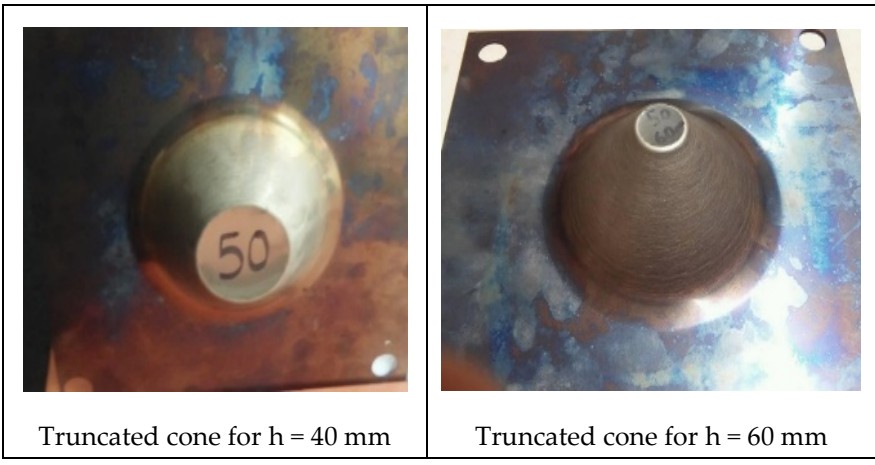

| Truncated cone for h = 40 mm | Truncated cone for h = 60 mm |

**Figure 7.** Experimental deformed truncated cone for $\alpha$ = 50°, h = 40 mm, and h = 60 mm.

### 3.1. Measurement of Deformed and Thickness Profile of the Deformed Truncated Cone

Four cutting planes (Plane 1, Plane 2, Plane 3, and Plane 4) were defined to perform a palpation and scan of the formed truncated cone surfaces (top and bottom), as shown in Figure 8. A measuring program is developed using Coordinate-Measuring Machine, which made it possible to carry out measurements on the upper surface and the inner surface of the truncated cone.

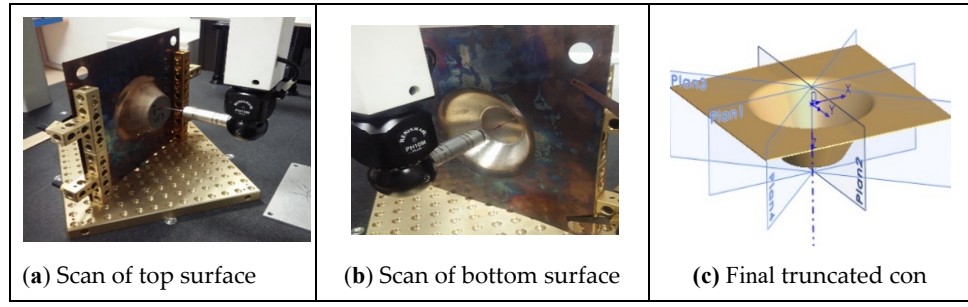

| (**a**) Scan of top surface | (**b**) Scan of bottom surface | (**c**) Final truncated con |

**Figure 8.** Scan of two truncated cone sides according to four cutting planes.

The distribution of the final thicknesses ($t_f$) and the profile of the deformed truncated cone according to the four cut planes makes it possible to evaluate the relative errors. Since the truncated cone has been asymmetrically observed, identical results should normally be obtained for all four planes. Figure 9 plots the final thickness distributions ($t_{plane1}$, $t_{plane2}$, $t_{plane3}$, $t_{plane4}$) and shows that this error is very small throughout the profile, and a very similar minimum thickness is obtained.

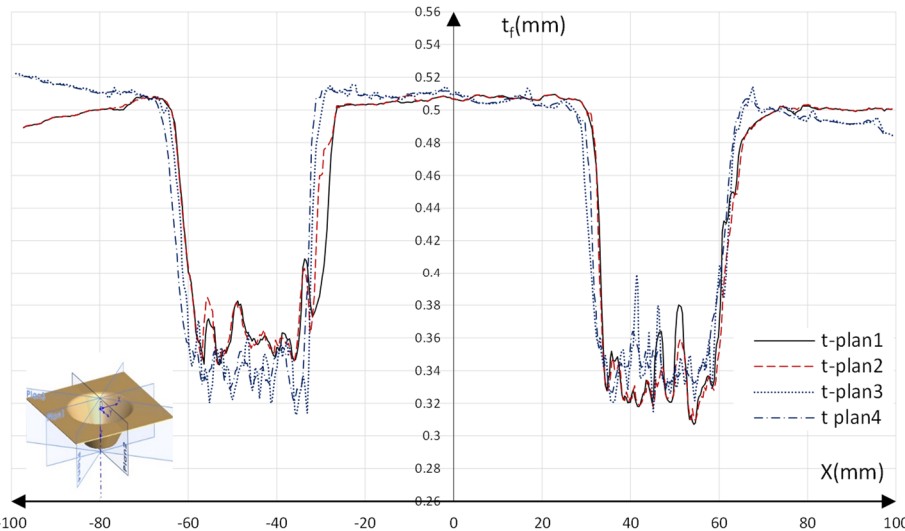

**Figure 9.** Experimental thicknesses (tf) according to the four cutting planes for h = 40 mm.

Experimental profile shapes according to the four cutting planes ($U_{plane1}$, $U_{plane2}$, $U_{plane3}$, $U_{plane4}$) are given in Figure 10, and relative errors of the profiles' shapes compared to the profile of plane 1 ($U_{plane1}$) are shown in Figure 11. It is clearly observed that errors are always small and do not exceed 3%. As shown in Figure 11, the error is significant at the tips of the cone entrance leaves called zone 1. This error decreases in zone 2 and then increases to cancel in the center of the bottom of the cone zone 3. Many of these errors are related to the palpation of the Coordinate-Measuring Machine.

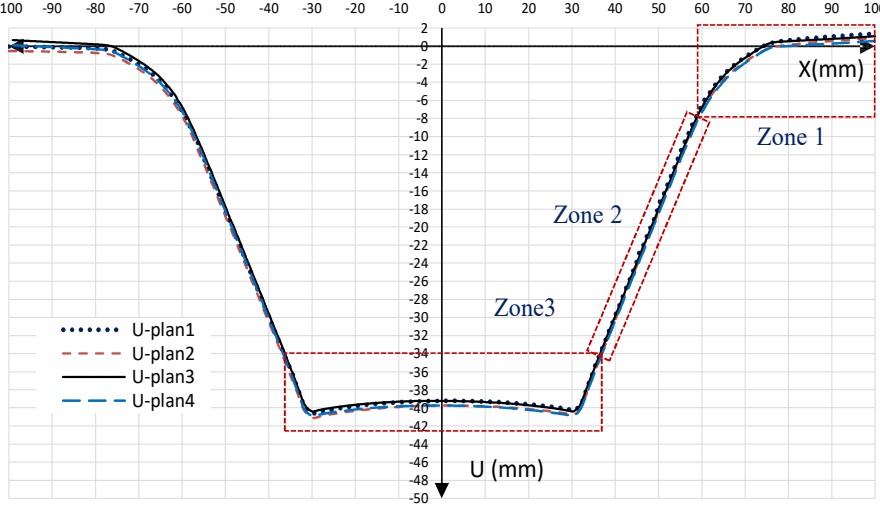

**Figure 10.** Experimental profiles shape according to the four cutting planes for h = 40 mm.

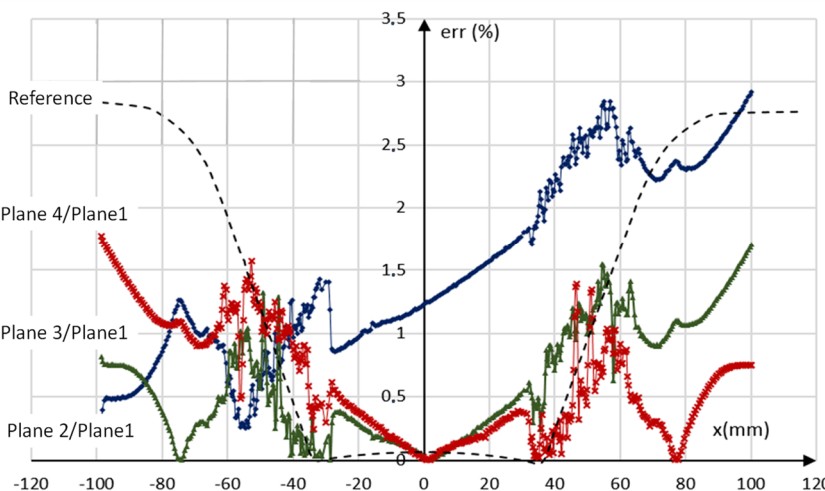

**Figure 11.** Relative profile error compared to the profile along to the plane 1 for h = 40 mm.

It was concluded that the errors are very small; only the profile along plane 1 will be kept for the presentation of the results in the rest of this paper. Therefore, the results that will be presented below for the thickness distribution or the measured profile will be in the cut plane (-X, X) as plane 1.

### 3.2. Predicted Profile Shape and Thickness Distribution of the Deformed Truncated Cone

For the FE simulation, results of the warm SPIF of truncated cone (h = 40 or h = 60 mm and $\alpha$ = 50°) are shown in the following Figure 12 in comparison with the experimental deformed shape. The thicknesses distributions ($t_f$) and profiles shapes (U) according to the four planes (Plane 1; Plane 2; Plane 3; Plane 4) are measured as shown in the following Figure 9.

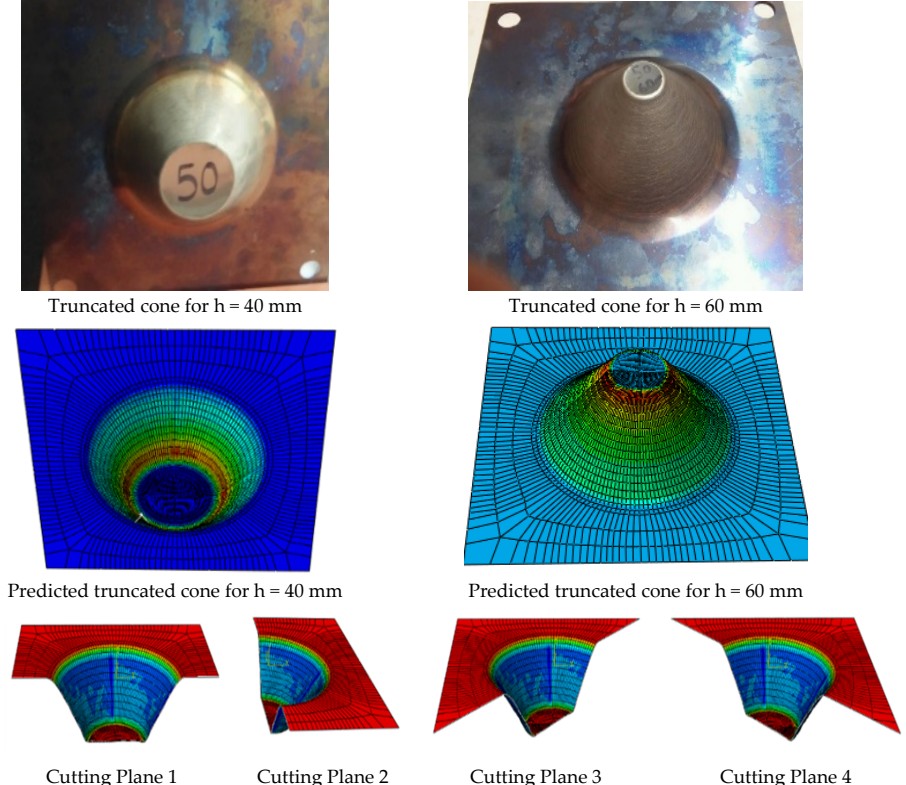

**Figure 12.** Four cutting planes measurement of FE simulation truncated cone.

The distribution of the predicted thicknesses according to the four planes ($t_{plane\ 1}$, $t_{plane\ 2}$, $t_{plane\ 3}$, $t_{plane\ 4}$) is plotted on the same graph to study the errors between the different curves. Figure 13 shows that these errors are very small throughout the distribution and gives very similar minimum thicknesses. The FE simulation results show that the formability of titanium alloys is improved for a temperature greater than 400 °C.

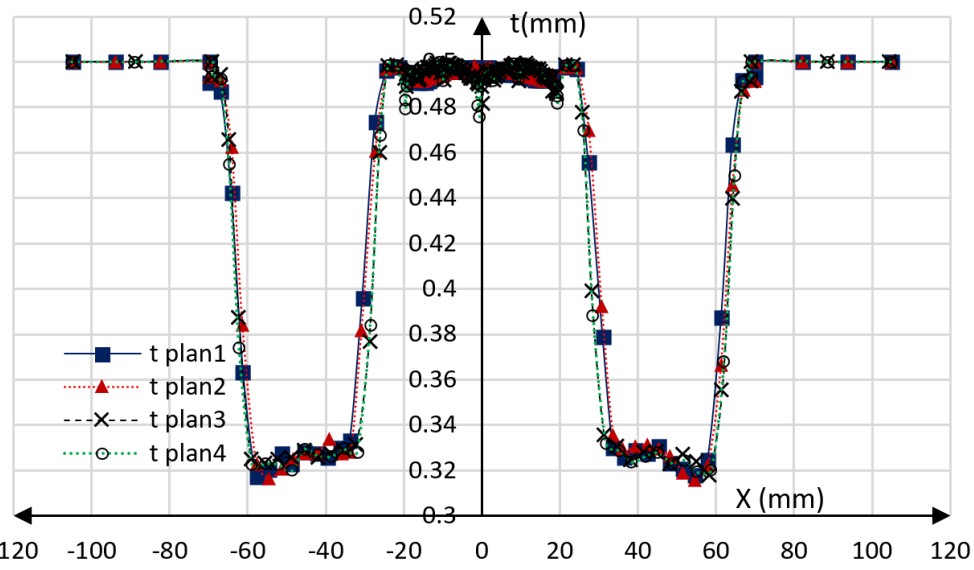

**Figure 13.** Predicted thicknesses distribution (tf) according to the four cutting planes for h = 40 mm.

For a temperature range of 450 °C, a truncated cone is obtained in dimensional compliance with the desired profile. The model considers the heating caused by the friction of the punch when it meets the blank. A fraction of the friction heat is driven to the sheet and localized, too, around the affected area. The temperature in this area increases as the angle of the simulated room increases [27]. The comparison of experimental and predicted results is presented below for the thickness distribution or the measured profile shape in the cut plane (−X, X).

To make a comparison between the predicted and experimental results, Figures 14 and 15 present the distribution of the thicknesses of truncated cones performed (h = 40 mm, h = 60 mm). The results of the thickness distribution for the case of a deep truncated cone (h = 60 mm) show a good agreement between the experimental and the numerical results. In addition, results show that heating improves the ductility of the material, and we manage to obtain a deep truncated cone up to h = 60 mm without breakage or defect in the folds of the sheet. The minimum thickness that we have reached is X = 31.5 mm. The reached minimum experimental final thickness is $t_{fmin}$ = 0.332 mm (Figure 13). On the other hand, the reached minimum thickness was predicted too, with FE simulation $t_{fmin}$ = 0.31 mm. The relative errors between these results are of the order of 7%, which is relatively small compared to the results obtained by M. Honapsheh et al. [33]. The thickness reduction is increasing with the depth and the minimum numerically thickness obtained for each depth is slightly greater than the experimentally measured value.

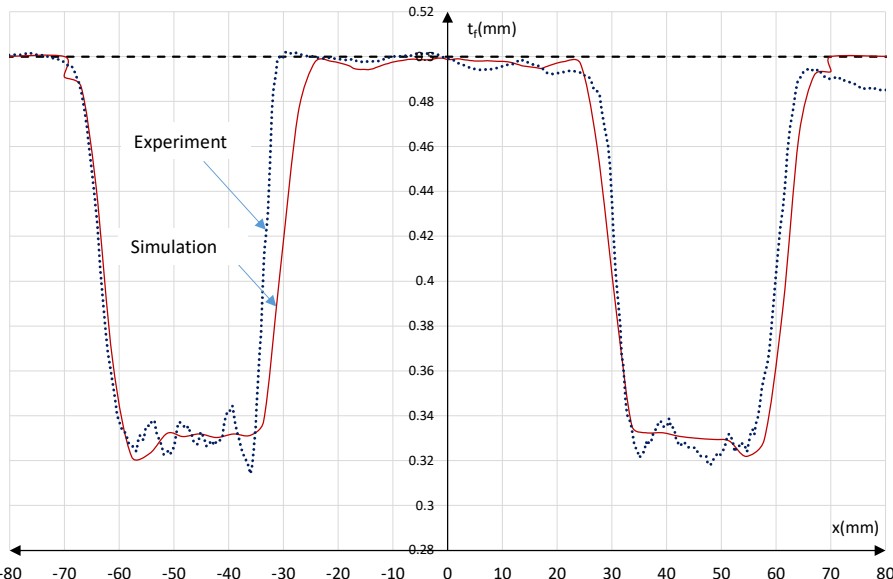

**Figure 14.** Experimental and FE numerical thickness distributions for depth h = 40 mm.

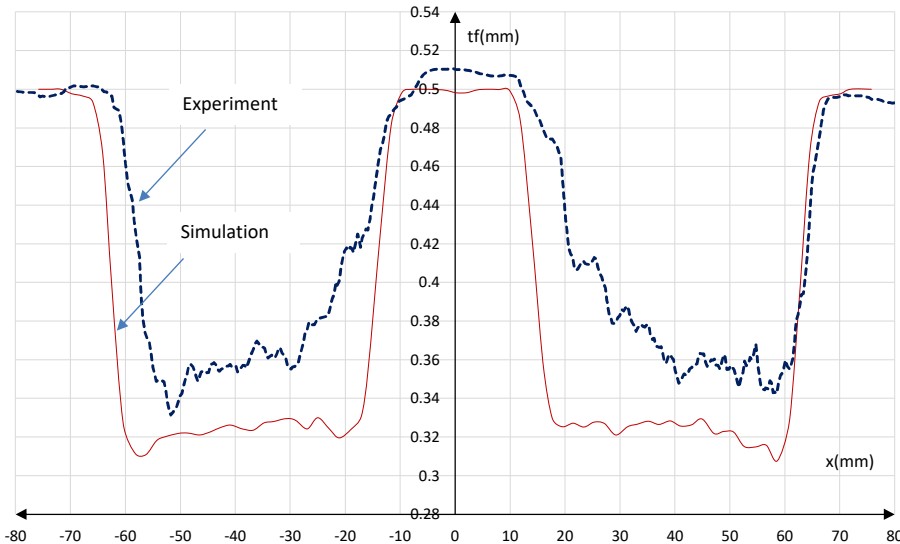

**Figure 15.** Experimental and FE numerical thickness distributions for depth h = 60 mm.

The obtained profiles shape for h = 60 mm (Figure 16) give correct shapes without warping often caused by heating. The uniformity of heating provided by the warm SPIF setup allows us to obtain parts in dimensional conformity. This is considered as a benefit of this type of heating, because often, we notice that the heating associated with the springback phenomenon causes an alteration in the final dimensions of the performed truncated cone. Dimensional conformity is firstly verified by the obtained profiles (Figure 16) and secondly by calculating the relative errors of the profiles compared to the real or theoretical profile presented in Table 5.

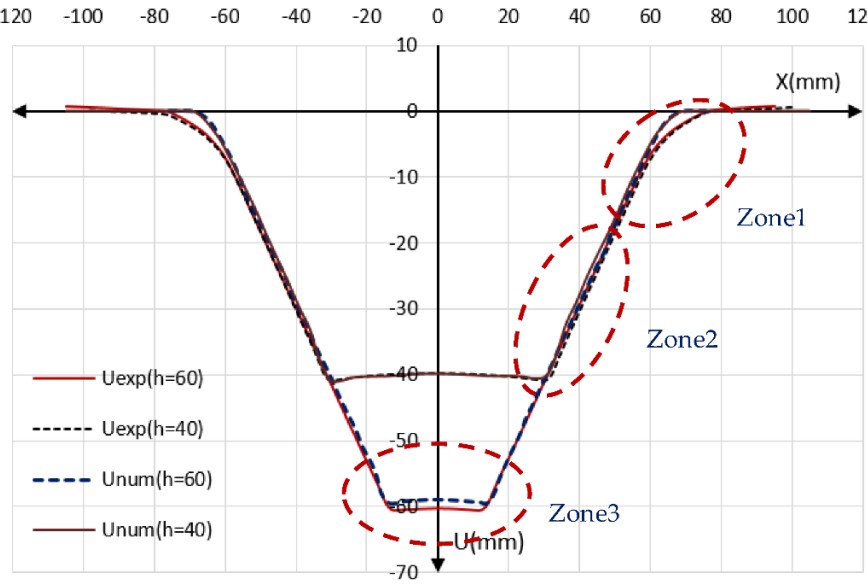

**Figure 16.** Experimental and FE numerical profile shapes for h = 60 m.

**Table 5.** Summary of experimental and numerical results.

| Depth (h) | 40 mm | 60 mm |
|---|---|---|
| Minimum predicted thickness: $t_{num}$ [mm] | 0.33 | 0.31 |
| FE minimum predicted thickness reduction ratio: $t_{num}/t_i$ [%] | 33.8% | 33.6% |
| Minimum experimental thickness: $t_{exp}$ [mm] | 0.33 | 0.30 |
| Experimental minimum thickness reduction ratio: $t_{exp}/t_i$ [%] | 34.4% | 38% |
| Error between the experimental and theoretical displacement in Zone 3 (bottom): $U_{exp}/U_{theo}$ [%] | 2.75% | 1% |
| Error between the FE simulation and theoretical displacement Zone 3 (bottom): $U_{num}/U_{theo}$ [%] | 3% | 3% |
| Error between the experimental and theoretical displacement in Zone 1 (leave level): $U_{exp}/U_{theo}$ [%] | 8.75% | 6% |
| Error between the FE simulation and theoretical displacement in Zone 1 (leave level): $U_{num}/U_{theo}$ [%] | 5.5% | 2% |

Figure 17 presents the experimental and numerical profile errors compared to the theoretical profile along plane 1 for h = 60 mm. As can been observed, errors for the profile are always less than 6%.

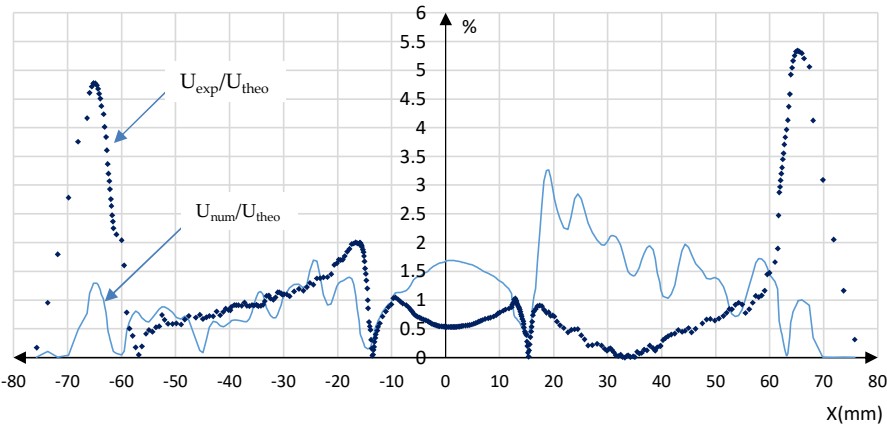

**Figure 17.** Relative error, $U_{exp}/U_{theo}$ and $U_{num}/U_{theo}$, for the case of (h = 60 mm, $\alpha = 50°$).

**Zone 1**: As shown in the summary in Table 5, of the comparison results, the major deviation observed in zone 1 (leave level) does not exceed 9% for the theoretical/experimental error at the cone leave level in the case of a truncated cone with a depth of h = 40 mm. This is an area placed between the clamping system and the first circle of the tool path (Figure 16). Large deformations, amplified by the heating, exist in this free zone due to the free part of the blank, which is affected by the punch at the start of forming.

**Zone 2**: It is the major deformed area of the truncated cone where the minimum value of the thickness is reached; the deviations are less than 2%. The numerical model gives an accurate prediction of the profile shape.

**Zone 3:** In the bottom of zone 3 of the truncated cone, a good agreement can be observed between the experimental and theoretical profile. The errors in this zone are less than 6%, which proves that the proposed numerical model allows good predictions of some phenomena.

The profile obtained by the prediction of the EF model, compared to the experimental profile shape in the cases of a deep truncated cone (h = 60 mm), gives a relatively satisfactory error of 4%. The numerical model makes it possible to predict the movements of the part and gives rise to a shape almost in dimensional conformity with that programmed path. The error at the level of leave is 6% (see Table 5). These errors could be improved by making modifications to the EF model in future work [33]. A comparison of the two profiles was made with different depths, which proves that the profile shape is not affected by the depth and the curves are in conformity [34].

## 4. Conclusions

In this paper, an experimental and numerical investigation on deep truncated cone accuracy improvement in a warm incremental sheet-forming process WSPIF of titanium alloy Ti-6Al-4V sheets was presented. The main advantages of this setup are its low cost and great simplicity, giving it the ability to be exploited in the industry. The experimental assays to form a deep truncated cone were performed at 450 °C with an experimental setup based on the use of heat cartridges.

The comparison of the experimental and numerical simulation results based on the thickness distribution and the final profile shape of the deep truncated cone showed the relevance of the numerical prediction and of the behavioral model used. The profile errors between numerical and experimental results do not exceed 4%. The thickness errors between simulation and measured parts are less than 7%. Through this study, it can be concluded that the proposed FE model can give an accurate prediction of the experimental results performed on the deep truncated cone of Ti-6Al-4V titanium alloys.

The results of the thickness reduction for a deep truncated cone (h = 60 mm) show a good approval for the hot forming of titanium alloys. This ensures that the heating improves the ductility of the material. Therefore, it is possible to make deep parts that can reach h = 60 mm without breaking the metal sheet. Errors between experimental and theoretical displacements are less than 6%, which prove that the profile shape is not affected by the depth.

However, larger errors have been observed between the experimental and theoretical profile for the depth of 40 mm. The improvement of the FE simulation tool and optimization of the tool path are perspectives of this work to minimize these errors.

**Author Contributions:** Investigation, B.S. and L.G.M.; Methodology, L.G.M. and A.C.; Project administration, L.G.M. and A.C.; Visualization, R.N.; Writing – original draft, B.S.; Writing – review & editing, L.G.M. and A.C. All authors have read and agreed to the published version of the manuscript.

**Funding:** The authors would like to acknowledge the valuable financial support of University of Technology of Troyes and Grand Est Region France.

**Data Availability Statement:** The data are taken from Saidi's doctoral thesis and are available for publication needs.

**Conflicts of Interest:** The authors declare no conflict of interest.

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
