# Peer review of "Accuracy and Sheet Thinning Improvement of Deep Titanium Alloy Part with Warm Incremental Sheet-Forming Process"

_jmmp, doi:10.3390/jmmp5040122_

Round 1

Reviewer 1 Report

This work presents an experimental procedure and numerical simulation to improve warn single point incremental sheet forming of deep truncated cone in Ti-6Al-4V 20.

The work needs some corrections before consideration for publication:

  1. Abstract: the methodology and the number of experiments as well as the results should be better explained in the abstract.
  2. Figure 1: Delete the center design as it is not needed. In the drawing on the right (section view) I recommend adding the identification of the installation location of the resistance electrical cartridge heaters, to make the reader easier to understand.
  3. Line 140: Replace “300x300mm²” with “300x300mm”, with space between number and unit of measure and without squaring.
  4. Line 141: Replace “70x70mm²” with “70x70mm”, with space between number and unit of measure and no square.
  5. Line 144: Use “S = 50 rpm” with space between number and unit of measure.
  6. Line 148: use the term “step down” instead of “depth increment”. I also recommend that the value of Δz = 0.5 mm be presented in the text, since in table 1 this value is constant.
  7. The wall angle (α) should be presented in section 2.1, as a constant value was selected.
  8. I recommend that in table 1 the parameters of all the experiments performed are presented. Add the column "no." with the number of each experiment and in column “h” present the values ​​used in each experiment.
  9. Figure 4 has the caption “T=450ºC", however in the photo it is written "440ºC". I recommend correction of the subtitle.
  10. Figure 5 is neither cited nor explained in the text. This explanation must be added in the methodology.
  11. Figure 7 and Figure 12 have the caption “Truncated cone for h=40mm”, however in the photo it is written “50”. I recommend correction of the subtitle.
  12. In lines 295 and 296 the following text is presented “However, the errors between numerical and experimental results are always less than 5%.”, however, in line 309 it is said that “does not exceed 9%”. Fix this.
  13. The Conclusion needs to be clearer and more objective. Say for example that profile errors do not exceed 3%; thickness errors between simulation and measured part less than x%; proves the profile shape is not affected by the depth.

Author Response

Recommandations

Answers

  1. Abstract: the methodology and the number of experiments as well as the results should be better explained in the abstract.

Corrected in red:

"two experiments with truncated cone having a depth of 40 and 60 mm"
"Small errors are observed between experimental and theoretical profiles. Moreover, errors between experimental and numerical displacements are less than 6%, which shows that the FE model gives accurate predictions for titanium alloy deep truncated cone."

  1. Figure 1: Delete the center design as it is not needed. In the drawing on the right (section view) I recommend adding the identification of the installation location of the resistance electrical cartridge heaters, to make the reader easier to understand.

This figure has been modified.

  1. Line 140: Replace “300x300mm²” with “300x300mm”, with space between number and unit of measure and without squaring.

Corrected in red

  1. Line 141: Replace “70x70mm²” with “70x70mm”, with space between number and unit of measure and no square.

Corrected in red

  1. Line 144: Use “S = 50 rpm” with space between number and unit of measure.

Corrected in red

  1. Line 148: use the term “step down” instead of “depth increment”. I also recommend that the value of Δz = 0.5 mm be presented in the text, since in table 1 this value is constant.

Corrected in red

  1. The wall angle (α) should be presented in section 2.1, as a constant value was selected.

Corrected in red:

"Reverse engineering approach for surface reconstruction of CAD models starting from 3D mesh data is performed to analyze the formability of deformed truncated cone with a wall angle α of 50°, using three-dimensional Coordinate-Measuring Machine."

  1. I recommend that in table 1 the parameters of all the experiments performed are presented. Add the column "no." with the number of each experiment and in column “h” present the values ​​used in each experiment.

Corrected in red

  1. Figure 4 has the caption “T=450ºC", however in the photo it is written "440ºC". I recommend correction of the subtitle.

Corrected in red

  1. Figure 5 is neither cited nor explained in the text. This explanation must be added in the methodology.

Corrected in red

  1. Figure 7 and Figure 12 have the caption “Truncated cone for h=40mm”, however in the photo it is written “50”. I recommend correction of the subtitle.

Corrected in red

  1. In lines 295 and 296 the following text is presented “However, the errors between numerical and experimental results are always less than 5%.”, however, in line 309 it is said that “does not exceed 9%”. Fix this.

Corrected in red

"Error between the experimental and theoretical displacements do not exceed 9%"

but

"Profile errors between numerical and experimental results do not exceed 4%. "

  1. The Conclusion needs to be clearer and more objective. Say for example that profile errors do not exceed 3%; thickness errors between simulation and measured part less than x%; proves the profile shape is not affected by the depth.

Corrected in red :

« Profile errors between numerical and experimental results do not exceed 4%. Thickness errors between simulation and measured parts is less than 7%. »

« Errors between experimental and theoretical displacements are less than 6%, which proves than the profile shape is not affected by the depth.

However, larger errors have been observed between the experimental and theoretical profile for the depth of 40 mm. The improvement of the FE simulation tool and optimization of the tool path are perspectives of this work to minimize these errors.

Reviewer 2 Report

The paper proposes a method of incremental stamping of titanium sheets. The method is based on the local heating of the workpiece  with the help of heating cartridges. The profile obtained and the thickness of the parts are compared with the values obtained by simulation.
I am of the opinion that the paper can be published after some improvements. The authors could present an image of the experimental stand. It could also be specified whether an experimental plan was followed. Was the influence of temperature or other parameters only monitored? There are typos that need to be corrected (for example, line 89 "warm" instead of "warn", in table 5, there are misspellings - "0., 328" or "8., 75").

Author Response

Recommandation

Answer

The authors could present an image of the experimental stand.

Figure 1 has been modified to present the experimental stand

It could also be specified whether an experimental plan was followed.

An experimental plan was used in a previous study [31] bus not in this one.

Was the influence of temperature or other parameters only monitored?

Influence of other parameters was presented in another paper [31]

 There are typos that need to be corrected (for example, line 89 "warm" instead of "warn", in table 5, there are misspellings - "0., 328" or "8., 75").

Corrected in blue : "warm"

table 5 : corrections in blue

Reviewer 3 Report

The paper presents a research regarding the accuracy and the sheet thinning obtained through warn single point incremental sheet forming implemented for frustum-of-cone parts from Ti-6Al-4V alloy. This topic is a hot research topic in ISF field (and is properly summarized in the introduction chapter of this paper) and it is already well known that the material heating improves the ductility properties, multiple solution being used for hard to form material heating. Even so, in this paper, the authors presents another heating solutions much more simple and easy to implement than others presented in the literature, a solution which suppose using cartridge heaters that are available on the marked and can be easy to be inserted in/on the sheet fixing device. Both, experimental and theoretical research were done and the results comparison shows that errors are always small and do not exceed a few percentages.

The paper is quite well structured in 4 chapters and several subsections. It has an introduction in the research topic with literature references regarding the method to increase formability for Ti-6Al-4V alloy. The second chapter present the experimental setup and the WSPIF process conditions and parameters, but also the theoretical study using FEM analysis in ABAQUS software together with the simulation setup and forming conditions. The research results are described in chapter three by using pictures for real manufactured parts, FEM simulated parts, tables and graphical representations for comparative results. There are analysed the dimensional accuracy and the thickness reduction obtained for both, the real manufactured parts and the virtual FEM deformed parts.

The conclusion chapter slight presents only several important aspects presented in the paper, but not as it should be in a conclusion chapter.

Thus, below are several recommendations for authors:

- in the paper there are followed two parts aspects, accuracy and sheet thinning. Thus, it is recommended to introduce both aspects in the paper title (now refers only on part accuracy);

- row 40 – “Incremental forming technology is developed to decrease shape errors in the conventional incremental forming process” – the phrase has no meaning. Maybe it was desired to write “Worm Incremental forming technology is developed to decrease shape errors in the conventional incremental forming process”;

- rows 41-53 – it is not necessary to make an incremental forming classification. At that Journal level, those thing are already well-known;

- maybe in the introduction it should be good to present also another improving accuracy methods (ex: forming using a circumferential hammering tool)

- row 104 – wrong expression “Experimental tests are performed is order to optimize the placement” should become “Experimental tests are performed in order to optimize the placement”;

- row 252 - wrong expression “We concluded…..” – is recommended “It was concluded……”. Never use first or third person pronoun when you write an article;

- several figures should be increased in order to be more visible and easier to be analysed (at least figures 9 and 10);

- rows 244-247 – this phrase it cannot be understood, I thing it needs rework - “The displacements measured final sheet profile according to the four cutting planes (Uplane1, Uplane2, Uplane3, Uplane4) given in Fig.10 show that the relative errors of the profiles shapes with respect to the first profile (Uplane1) were given in Fig.11, on which the profile (Uplane1) is associated to provide an evolution order of the errors in each zone;

- row 331 – why table 5 does not contain also the results for zone 2 ? Also in table 5, some values has the “%” symbol, others does not contain this symbol. The value 8,.75 should be 8.75, without comma “,”. Please rework.

- it wold be interesting if will be presented comparative results and graphical representation for parts manufactured using different heat temperatures (between 400°C and 600°C);

- the results presented in figures 15, 16 and 17 should be discussed / commented in text from chapter 3 (at least some brief comments about those comparisons);

- in the conclusion chapter should be clearly specified which is the novelty of the presented methodology in comparison with other heating method which is common in the industry. The conclusion does not present which are the benefits of the researched method, which are the improvements regarding the accuracy or the sheet thinning. The conclusion chapter is too superficial and should be improved;

- future research work should be presented at the end of the paper.

Author Response

Recommandations

Answers

in the paper there are followed two parts aspects, accuracy and sheet thinning. Thus, it is recommended to introduce both aspects in the paper title (now refers only on part accuracy);

the title has been changed : Accuracy and sheet thinning improvement of deep titanium alloy part with warm incremental sheet forming process

- row 40 – “Incremental forming technology is developed to decrease shape errors in the conventional incremental forming process” – the phrase has no meaning. Maybe it was desired to write “Worm Incremental forming technology is developed to decrease shape errors in the conventional incremental forming process”;

corrected in purple :

Several techniques have been developed to decrease shape errors in the conventional incremental forming process: warm incremental forming technology, optimization of the process parameters, tool path optimization, use of circumferential hammering tool…

- rows 41-53 – it is not necessary to make an incremental forming classification. At that Journal level, those thing are already well-known;

it is a reminder of the two categories of process.

- maybe in the introduction it should be good to present also another improving accuracy methods (ex: forming using a circumferential hammering tool)

Added in purple :

Several techniques have been developed to decrease shape errors in the conventional incremental forming process: warm incremental forming technology, optimization of the process parameters, tool path optimization, use of circumferential hammering tool…

- row 104 – wrong expression “Experimental tests are performed is order to optimize the placement” should become “Experimental tests are performed in order to optimize the placement”;

corrected in purple :

- row 252 - wrong expression “We concluded…..” – is recommended “It was concluded……”. Never use first or third person pronoun when you write an article;

corrected in purple :

It was concluded that the errors are very small, only the profile along the plane 1 will be kept for the presentation of the results in the rest of this paper.

- several figures should be increased in order to be more visible and easier to be analysed (at least figures 9 and 10);

they have been increased

- rows 244-247 – this phrase it cannot be understood, I thing it needs rework - “The displacements measured final sheet profile according to the four cutting planes (Uplane1, Uplane2, Uplane3, Uplane4) given in Fig.10 show that the relative errors of the profiles shapes with respect to the first profile (Uplane1) were given in Fig.11, on which the profile (Uplane1) is associated to provide an evolution order of the errors in each zone;

corrected in purple :

Experimental profiles shapes according to the four cutting planes (Uplane1, Uplane2, Uplane3, Uplane4) are given in Fig.10 and relative errors of the profiles shapes compared to the profile of plane 1 (Uplane1) are shown in Fig.11.

- row 331 – why table 5 does not contain also the results for zone 2 ? Also in table 5, some values has the “%” symbol, others does not contain this symbol. The value 8,.75 should be 8.75, without comma “,”. Please rework.

- it wold be interesting if will be presented comparative results and graphical representation for parts manufactured using different heat temperatures (between 400°C and 600°C);

These experimentations have not been performed for lack of titanium parts. In addition, experimental tests with titanium alloy are generally presented for temperatures between 400 and 500°C in the literature. The temperature chosen was consistent with the state of the art and our experimental tests.

- the results presented in figures 15, 16 and 17 should be discussed / commented in text from chapter 3 (at least some brief comments about those comparisons);

corrected in purple

Figures are discussed.

"Fig. 17 presents the experimental and numerical profile errors compared to the theoretical profile along to the plane 1 for h = 60 mm. As can been observed, errors are always less than 6%."

- in the conclusion chapter should be clearly specified which is the novelty of the presented methodology in comparison with other heating method which is common in the industry. The conclusion does not present which are the benefits of the researched method, which are the improvements regarding the accuracy or the sheet thinning. The conclusion chapter is too superficial and should be improved;

The conclusion chapter has been improved:

"The main advantages of this setup are its low cost, great simplicity, giving it the ability to be exploited in the industry. "

"Profile errors between numerical and experimental results do not exceed 4%. Thickness errors between simulation and measured parts is less than 7%."

"Errors between experimental and theoretical displacements are less than 6%, which proves than the profile shape is not affected by the depth."

"However, larger errors have been observed between the experimental and theoretical profile for the depth of 40 mm. "

- future research work should be presented at the end of the paper.

Corrected in red
